# Effects of Acute Stress on Metabolic Interactions Related to the Tricarboxylic Acid (TCA) Cycle in the Left Hippocampus of Mice

**DOI:** 10.3390/metabo14120699

**Published:** 2024-12-11

**Authors:** Chang-Soo Yun, Yoon Ho Hwang, Jehyeong Yeon, Hyeon-Man Baek, Dong Youn Kim, Bong Soo Han

**Affiliations:** 1Department of Radiation Convergence Engineering, College of Software and Digital Healthcare Convergence, Yonsei University, 1, Yeonsedae-gil, Heungeop-myeon, Wonju 26493, Republic of Korea; ycs0709@yonsei.ac.kr (C.-S.Y.);; 2Institute for Human Genomic Study, College of Medicine, Korea University, Seoul 15355, Republic of Korea; 3Department of Health Sciences and Technology, Gachon Advanced Institute for Health Sciences and Technology (GAIHST), Gachon University, Incheon 21999, Republic of Korea; 4Department of Biomedical Engineering, College of Software and Digital Healthcare Convergence, Yonsei University, 1, Yeonsedae-gil, Heungeop-myeon, Wonju 26493, Republic of Korea

**Keywords:** acute stress response, proton magnetic resonance spectroscopy, metabolites interaction

## Abstract

Background/objectives: The acute stress response affects brain metabolites closely linked to the tricarboxylic acid (TCA) cycle. This response involves time-dependent changes in hormones and neurotransmitters, which contribute to resilience and the ability to adapt to acute stress while maintaining homeostasis. This physiological mechanism of metabolic dynamics, combined with time-series analysis, has prompted the development of new methods to observe the relationship between TCA cycle-related brain metabolites. This study aimed to observe the acute stress response through metabolic interactions using time-series proton magnetic resonance spectroscopy (1H-MRS) in the left hippocampus of mice. Methods: In this study, 4-week-old male C57BL/6N mice (*n* = 24) were divided into control (*n* = 12) and acute stress groups (*n* = 12). Acute stress was induced through a 2 h restraint protocol. Time-series 1H-MRS data were obtained on the left hippocampus of both groups using a 9.4 T 1H-MRS scanner. Time-series MRS data were quantified using LCModel, and significant metabolic interactions were identified through Spearman correlation analysis, a one-tailed sign test, and false discovery rate correction. Results: No significant metabolic correlation coefficient was observed in the control group. However, in the acute stress group, glutathione (GSH) and N-acetyl aspartate (NAA) showed a significant positive correlation over time, with a high correlation coefficient exceeding 0.5. Conclusions: Temporal measurement of GSH and NAA, combined with correlation analysis, offers a comprehensive understanding for the metabolic dynamics during acute stress. This approach emphasizes their distinct roles and interdependence in the progression of oxidative stress, mitochondrial function, and the maintenance of physiological homeostasis.

## 1. Introduction

In modern society, stress not only affects social interactions but also disrupts neurological homeostasis in the human brain. Prolonged exposure to stress can impair neurochemical processes and the ability to adapt to external stimuli [1,2]. Among various stressors, short-term, intense stimuli—such as physical, psychological, and social stressors—are classified as acute stress [3]. Acute stress immediately influences neurochemical processes, and repeated exposure can contribute to the development of chronic mental disorders or post-traumatic stress disorder [4]. Therefore, studying the effects of acute stress is crucial for maintaining neurochemical homeostasis and preventing the onset of mental disorders.

Acute stress impacts the hypothalamic–pituitary–adrenal (HPA) axis, which responds to external stimuli through the release of corticotropin-releasing hormone (CRH) from the hypothalamus [5,6]. CRH stimulates the pituitary gland to secrete adrenocorticotropic hormone (ACTH), which in turn activates the adrenal glands to produce corticosterone (CORT). CORT travels through the bloodstream and binds to glucocorticoid receptors (GR) in the hippocampus, affecting neurotransmitter metabolites, such as glutamate (Glu), glutamine (Gln), and gamma-aminobutyric acid (GABA) [5,6,7,8,9]. Consequently, acute stress can lead to reduced antioxidant enzyme activity and increased neurotoxicity due to excessive Glu release [10]. Glu interacts with alpha-ketoglutarate (α-KG) in the tricarboxylic acid (TCA) cycle to maintain physiological homeostasis [11]. The TCA cycle, which involves α-KG, succinate, fumarate, and oxaloacetate, plays a key role in adenosine triphosphate (ATP) production and metabolic interactions [12]. The aspartate system interacts with fumarate in the TCA cycle to help maintain energy homeostasis [11,12,13]. Thus, monitoring brain metabolites in the glutamatergic and aspartate systems is crucial for understanding the effects of acute stress [14,15,16,17].

Proton magnetic resonance spectroscopy (1H-MRS) is a non-invasive technique used to quantify brain metabolite concentrations. 1H-MRS has been extensively employed to investigate the effects of acute stress on brain metabolite levels. According to Kim et al., Glu levels increase in the cerebral cortex and hippocampus following acute stress [18]. Additionally, N-acetyl aspartate (NAA) levels decrease in the hippocampus and cingulate cortex under fear conditions [19]. These studies demonstrate that the effects of acute stress can be observed in the glutamatergic and aspartate systems at the level of brain metabolites. The previous studies relied on endpoint analysis to provide information on whether specific metabolites are affected by acute stress response [18,19,20]. However, they do not offer insights into the interactions between the brain metabolites involved in the physiological response to acute stress.

Acute stress induces physiological variations in the brain that evolve over time, encompassing both short- and long-term effects, such as hormonal changes and structural and functional alterations, including dendritic retraction and Glu release [21,22]. These prolonged changes are particularly relevant in stress-related in stress-related disorders. Thus, it is necessary to investigate how these metabolic interactions change over time in response to acute stress [22]. To address this, we obtained time-series 1H-MRS data for 2 h and found significant differences in alanine (Ala), Glu, and NAA levels between the two groups [23]. Also, we reported that the control group maintained a small-world network regardless of time, unlike the acute stress group [24]. Although previous studies have demonstrated an acute stress response over time, observing the physiological interactions between brain metabolism during this response remains challenging. Understanding metabolic interactions over time is important for identifying the relationship between physiological homeostasis and the acute stress response.

This study aimed to observe the acute stress response through metabolic interactions over time using time-series 1H-MRS data. We analyzed the interactions of five metabolites in the hippocampus of mice: Gln, Glu, glutathione (GSH), GABA, and NAA involved in the glutamatergic and aspartate systems, which are closely related to the TCA cycle.

## 2. Materials and Methods

### 2.1. Experimental Animals

All experimental protocols were approved by the Institutional Animal Care and Use Committee and conducted at the Lee Gil Ya Cancer and Diabetes Institute, which is accredited by the Association for Assessment and Accreditation for Laboratory Animal Care International (AAALAC). AAALAC accredits research centers worldwide that the organization has found to follow the international legal standards and achieve excellence in animal care and use. All experimental procedures complied with the Center of Animal Care and Use (CACU) for Animal Research (Guidelines for Animal Users) and were conducted following the ARRIVE guidelines [25].

C57BL/6N mice (ORIENT BIO Inc., Seongnam-si, Gyeonggi-do, Republic of Korea, ages = 4 weeks, body weight = 18–25 g, sex = male, *n* = 24) were used as experimental animals. Three mice were housed in transparent cages for 2 weeks to acclimatize them to the experimental environment. The mice were not constrained to consume water and food under controlled conditions (temperature = 21–23 °C, humidity = 50–60%) with a 12 h/12 h light and dark cycle (lights on from 06:00 to 18:00). Mice were randomly divided into the control (*n* = 12) and acute stress (*n* = 12) groups. Mice in the acute stress group were exposed to restraint stress for 2 h in a 50 mL plastic conical tube (3 cm in diameter and 10 cm in length), following the stress protocol described in previous studies [23,24,25,26,27]. The plastic tube had holes near the body (3 mm in diameter) and the mouth (5 mm in diameter) of the mice to promote ventilation and airflow during the procedure. To ensure that the mice were immobilized completely, the residual space in the tube was blocked and filled with material to pressurize and secure the body and head of the mouse. The restraint stress procedure was conducted in an air-conditioned room under background noise to maintain consistency in the experimental environment. Immediately after the 2 h restraint stress protocol was completed, the mice were used for time-series MRS data acquisition. Mice in the control group were not exposed to restraint stress and remained undisturbed in their cages until the beginning of data acquisition.

### 2.2. Proton Magnetic Resonance Spectroscopy Protocol

All MRS data were acquired using a 9.4 T Bruker BioSpec Avance 3 94/20 USR (Bruker BioSpin GmbH, Ettlingen, Germany) at the Core-Facility for Cell to In Vivo Imaging. T2-weighted images were acquired using rapid acquisition with relaxation enhancement (RARE) with the following parameters: TR = 5000 ms, TE = 48 ms, field of view = 3 × 3 cm^2^, matrix size = 256 × 256, slice thickness = 1 mm, and volume of interest (VOI) = 1.8 × 3.4 × 1.8 mm^3^, and 11.016 μL. The VOI was located in the left hippocampus, which is closely related to acute stress. A fast automatic shimming technique by mapping along projections (FASTMAP) was used to improve the homogeneity of the magnetic field in the VOI. This was performed repeatedly to fit the linewidth of the water into the same VOI under 14 Hz. Single-voxel 1H-MRS data were acquired using a point-resolved spectroscopy sequence with the following acquisition parameters: TR = 4000 ms, TE = 10 ms, number of excitations = 512, complex points = 4096, and spectral width = 5000 Hz. The variable-power RF pulses were used for signal suppression outside the VOI. The unsuppressed water signal was obtained to perform eddy current correction to obtain the absolute quantification of the metabolites.

All MRS data were acquired every 34 min 8 s, four times in total, in both experimental groups. Mice were anesthetized with isoflurane (starting at 4.0% and maintained at 1.0–2.0% during the experiment) in a 1:2 O_2_–air mixture that was delivered to the mouth using an anesthesia apparatus. Mice were placed on a flat mouse bed in the prone position and tightly fixed with a bite bar and two ear inserts. A water-heated body warming system was used to maintain a stable body temperature of 38 °C during anesthesia. Respiration rate was continuously monitored using MR-compatible instruments (SA Instruments, Inc., Stony Brook, NY, USA) to adjust the anesthetic concentration to the mouse’s condition while time-series MRS data were acquired. Figure 1A presents the experimental protocol for both mouse groups.

### 2.3. Metabolite Quantification

To minimize the magnetic inhomogeneity due to the long scan time, we performed a frequency and phase shift correction of 512 free induction decay (FID) signals. The reference signal was selected using the mean squared error, which had the smallest difference between each 512 FID signal. FID-A software [Version 1.2], which is based on MATLAB [R2020a, Natick, MA, USA: MathWorks Inc.], was used to perform the phase and frequency shifts between the reference FID signal and other FID signals [28,29]. After performing phase and frequency correction, the 512 FID signal data were divided into 2 256 FID data points to improve the temporal resolution of the brain metabolite fluctuations (Figure 1B). Finally, 8 time points MRS data were used per group (control and acute stress).

The LCModel was used to quantify the MRS data based on the stimulated basis-set, which consisted of 17 metabolites. In this study, 11 metabolites, including Ala, creatine, phosphocreatine, GABA, Glu, Gln, phosphorylcholine, GSH, myo-inositol, NAA, and taurine, satisfied the Cramer–Rao lower bound (CRLB) of <35% at all time points. To investigate the acute stress effect of TCA cycle-related brain metabolites, we sorted the five brain metabolites in the left hippocampus, which included the glutamatergic and aspartate system: GABA, Gln, Glu, GSH, and NAA. One mouse was excluded from each group in the analysis because their data did not meet the reliability criterion of CRLB < 35%.

### 2.4. Statistical Analysis

The Spearman correlation coefficients between brain metabolites were calculated using the time-series MRS data for each mouse. For each group, a non-parametric one-tailed sign-rank test was performed on the metabolic correlation coefficients to evaluate their significance. To enhance the reliability of the results, false discovery rate (FDR) correction was applied to the *p*-values obtained from the sign-rank test for each group. To select pairs of metabolites with strong associations, we selected pairs of metabolites with a correlation coefficient greater than 0.5. Metabolite pairs with correlation coefficients having adjusted *p*-values greater than 0.05 were considered statistically significant [25]. Figure 2 illustrates the method for calculating metabolic correlation matrices for each group and demonstrates how the significant Spearman correlation coefficients between brain metabolites are compared between the control and acute stress groups.

## 3. Results

This study observed the acute stress response through metabolic interactions in the mouse hippocampus using time-series MRS data. The correlation coefficients between brain metabolites provide insights into the physiological metabolic interactions that adapt to acute stress responses. Table 1 represents the mean concentrations (±standard error) of the metabolites (GABA, Gln, Glu, GSH, and NAA) across eight time points for control and acute stress groups.

Table 2 presents the correlation coefficients and the adjusted *p*-values (at a significance level of 0.05) between metabolites related to the TCA cycle in both experimental groups. No significant correlations were observed in the control group. However, in the acute stress group, significant metabolic correlations were found for Gln–Glu, Gln–NAA, and GSH–NAA, with only the GSH–NAA pair exhibiting a high positive correlation coefficient greater than 0.5.

The high correlation between GSH and NAA in the acute stress group is illustrated in Figure 3, which depicts the temporal changes in the mean concentrations of GSH and NAA for both experimental groups. For GSH, the control group exhibited an approximately monotonic increase, ranging from 1.670 to 1.895 (Figure 3A). In contrast, the acute stress group displayed a slight increase from 1.661 to 1.695 up to time point 4, followed by a significant rise to 1.828 after time point 4 (Figure 3C). For NAA, the control group exhibited a slight decrease in mean concentration from 6.149 to 6.074 up to time point 3, followed by a monotonic increase to 6.478 after time point 3 (Figure 3B). In the acute stress group, the mean concentration of NAA showed a pattern similar to that of the control group up to time point 4, with slightly lower values than the control group, although the difference was not statistically significant. From time point 5 onward, the mean remained nearly constant (Figure 3D). Compared to the control group, the mean concentrations of GSH and NAA in the acute stress group exhibited similar temporal patterns (Figure 3C,D). Until time point 4, there were slight changes in the mean concentrations, which remained lower than those of the control group. From time point 5 onward, the concentrations increased, becoming comparable to those of the control group, with minimal or no changes over time. Figure 4 shows the correlation between GSH and NAA in the mice hippocampus between the control and acute stress groups. The acute stress group showed a strong positive correlation between GSH and NAA, indicating that changes in GSH levels had a more pronounced impact on NAA levels compared to the control group.

## 4. Discussion

This study investigated the hippocampus of mice subjected to acute restraint stress using 1H-MRS. Temporal changes in the concentrations of five TCA cycle-related metabolites—GABA, Gln, Glu, GSH, and NAA—were analyzed. Metabolite pairs with statistically significant, high correlation coefficients (>0.5) were identified, and the relationships between these highly correlated metabolite pairs and their roles in the TCA cycle during the stress response were explored.

Brain metabolites are intricately interconnected to support proper synaptic transmission and neural activity [30]. Acute stress increases the metabolic demands of cells within the central nervous system, where rapid adaptation is crucial for maintaining homeostasis and protecting against potential damage. It also significantly affects neurochemical homeostasis, particularly by influencing metabolic interactions within the glutamatergic and aspartate metabolic systems [15,16,17,18,19]. The TCA cycle plays an important role in alleviating acute stress through various interconnected mechanisms that enhance cellular and systemic resilience. It provides energy, regulating neurotransmitter balance, enhancing antioxidant defenses and preserving mitochondrial integrity, and maintaining cellular homeostasis [31,32,33,34,35,36,37,38]. Moreover, a statistically significant and strong correlation was observed between the temporal changes in GSH and NAA levels in the acute stress group (Figure 4). This finding underscores a functional interdependence between the two metabolites in responding to stress.

In this study, the mice in the acute stress group were subjected to 2 h of restraint stress, followed by temporal measurements of GSH and NAA levels using 1H-MRS. Figure 3C,D shows that during the early stages following restraint stress, both GSH and NAA concentrations in the hippocampus were significantly lower than those in the control group. This reduction likely reflects acute oxidative stress and mitochondrial dysfunction, impairing TCA cycle activity and energy metabolism. However, during the later stages, the concentrations of GSH and NAA returned to levels comparable to those of the control mice, indicating a recovery in mitochondrial function and cellular metabolism. Moreover, a statistically significant and strong correlation was observed between the temporal changes in GSH and NAA levels in acute stress group (Figure 4). This finding underscores a functional interdependence between the two metabolites in responding to stress.

GSH, a tripeptide composed of Glu, cysteine, and glycine, is an important component of the cellular defense system. It is rapidly synthesized and consumed in response to stress [39,40,41,42]. GSH, a critical antioxidant, plays a key role in neutralizing reactive oxygen species (ROS), which are overproduced during acute stress and can disrupt mitochondrial enzymes involved in the TCA cycle. The initial decrease in GSH following the acute stress likely corresponds to its consumption in neutralizing ROS which can impair mitochondrial enzymes critical for the TCA cycle, reducing energy production [43]. The recovery of GSH levels after time point 4 suggests the activation of compensatory mechanisms, such as increased GSH synthesis or recycling, enabling the restoration of redox balance and stabilization of the TCA cycle [44]. Similarly, NAA, as a marker of neuronal integrity and activity associated with mitochondrial energy metabolism, is sensitive to changes in neuronal mitochondrial function and energy demands [45,46,47]. It is primarily synthesized in neurons, where it is formed by combining aspartate and acetyl-CoA in the mitochondria [45]. The initial reduction in the NAA levels reflects mitochondrial dysfunction and impaired neuronal energy metabolism under stress [47,48,49]. The subsequent recovery of NAA levels indicates restoration of the TCA cycle function and mitochondrial activity, essential for neuronal metabolic homeostasis [50,51,52,53,54]. The measurements of GSH and NAA over time enable us to observe dynamic changes in metabolic responses, providing insights into the temporal patterns of adaptation during acute stress [55,56,57,58,59,60]. Unlike single time-point measurements, time-series analysis allows us to observe the progression and recovery of oxidative stress and mitochondrial function, revealing critical phases of metabolic shifts.

This study had several limitations. First, the sample size was insufficient to meet normality requirements and establish definitive metabolic interactions between brain metabolites. To address this, we employed the sign test for statistical analysis and applied FDR correction to enhance data reliability. Second, the time resolution, with approximately 17 min intervals between measurements, was inadequate to observe real-time metabolic dynamics in the brain. Therefore, the correlation coefficients between metabolites in this study reflect the correlation between time-series data of cumulative concentrations during the 17 min stress response. Third, we did not measure N-acetylcysteine (NAC), a precursor of GSH, which is directly involved in stress response and glutathione metabolism because measurement of NAC is challenging due to its low concentration and spectral overlap with other N-acetyl compounds, such as NAA [46]. NAC measurement could provide a more comprehensive understanding of the metabolic response to acute stress. Fourth, we did not assess other stress-related markers, such as total ROS or oxidative stress indicators (e.g., malondialdehyde or superoxide dismutase), which provide the precursor of the stress response. These markers would clarify the physiological stress levels between groups. Despite this, our focus on GSH-related metabolites provided significant insights into the stress response mechanisms. Further studies combining 1H-MRS and oxidative stress markers may help to increase our understanding of stress-induced metabolic interactions following acute stress. Lastly, we used isoflurane-based inhalation anesthesia to minimize the accumulation of anesthetic metabolites and ensure high reproducibility. While it is necessary to acquire 1H-MRS data with minimal movement and stress, anesthesia itself could have influenced the metabolic measurements. However, isoflurane has been shown to have minimal long-term effects, allowing for up to 8 h of anesthesia without significant differences [61,62]. Despite these limitations, this study successfully provided a non-invasive measurement of the dynamic changes in metabolite interactions following an acute stress response.

## 5. Conclusions

This study aimed to observe the acute stress response using brain metabolic interactions through correlation coefficients, which are closely related to the TCA cycle, using time-series 1H-MRS data. The combined advantages of measuring GSH and NAA over time and analyzing their correlation lie in their ability to provide a comprehensive understanding of metabolic dynamics and interactions during acute stress. Temporal measurement captures the progression of oxidative stress and mitochondrial function, offering insights into critical phases of metabolic adaptation and recovery. This approach reveals not only the individual roles of GSH and NAA but also their interdependence in maintaining physiological homeostasis.

## Figures and Tables

**Figure 1 metabolites-14-00699-f001:**
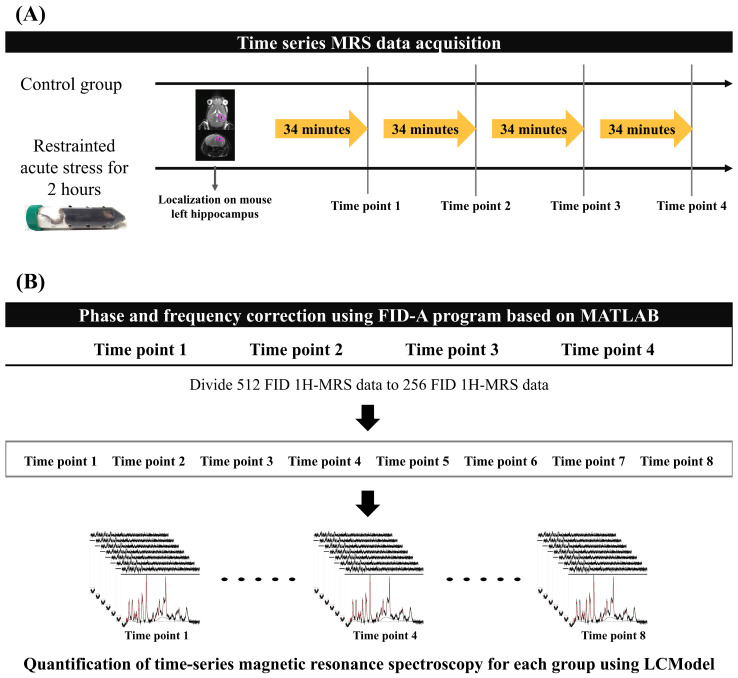
Acquisition and quantification protocol for MRS data: (**A**) Time-series MRS data were obtained for each group without time intervals. The region of interest indicates the left hippocampus of the mouse, where localization for MRS data acquisition was performed. (**B**) The 512 FID MRS data were conducted to phase and frequency correction using the FID-A program to reduce the magnetic inhomogeneity effect. After FID correction of the time-series MRS data, we divided the 256 FID data to increase the time resolution of each mouse brain metabolite to calculate metabolite interactions. The black lines represent the baseline spectra in the MRS data, and the red lines indicate the quantified metabolites. FID, free induction decay; MRS, magnetic resonance spectroscopy.

**Figure 2 metabolites-14-00699-f002:**
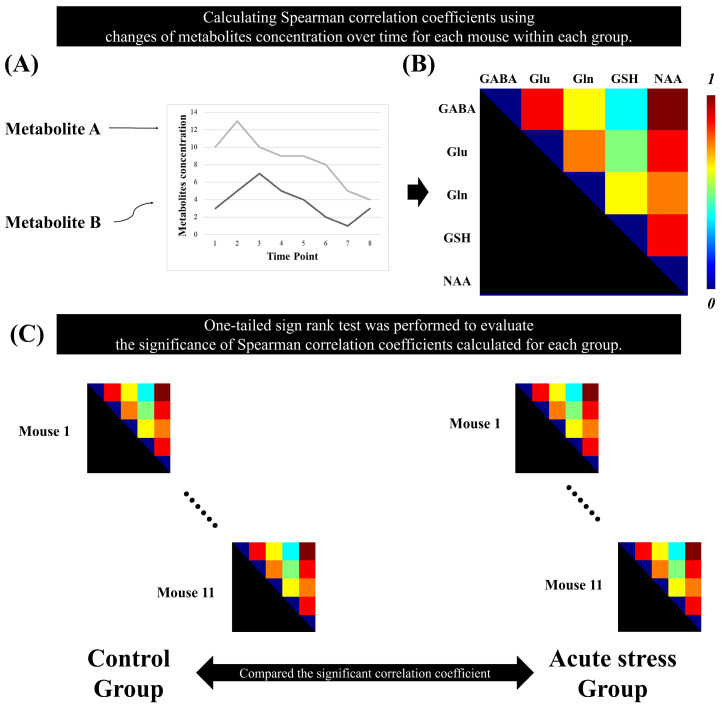
Statistical analysis to identify the significant metabolic correlation coefficients in the time-series magnetic resonance spectroscopy data. (**A**) represents the quantification of metabolite concentrations over time. The plot displays changes in the concentrations of Metabolite A and Metabolite B across multiple time points. (**B**) Spearman’s correlation coefficient was calculated between the metabolites over time for each mouse. (**C**) A one-tailed signed-rank test was used to determine significant correlation coefficients in each group.

**Figure 3 metabolites-14-00699-f003:**
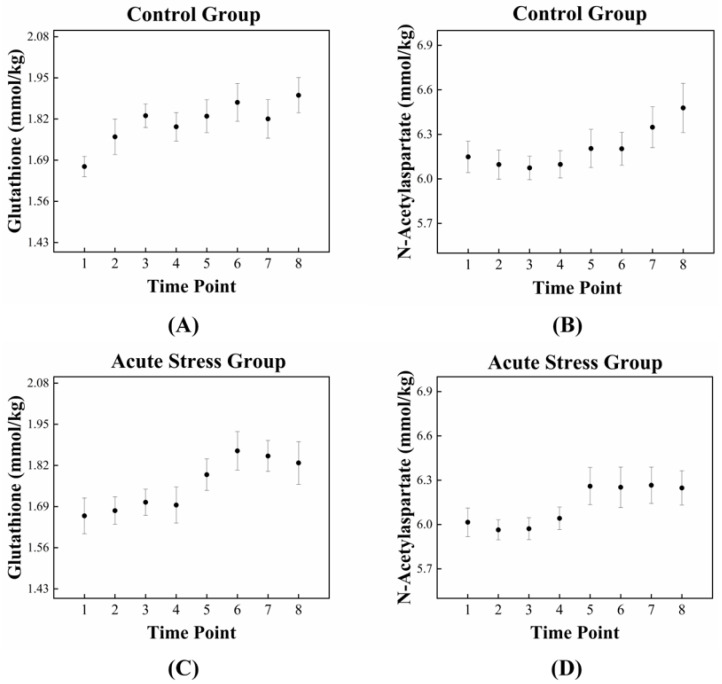
Dynamic fluctuations in GSH and NAA in the control and acute stress groups over time. In the control group (**A**), GSH levels are initially higher compared to those in the acute stress group (**C**). However, GSH levels in the acute stress group begin to increase after 85 min (time point 5), eventually reaching levels similar to those of the control group. In the control group (**B**), NAA levels gradually increased over time, whereas in the acute stress group (**D**), NAA levels rapidly increased after 85 min (time point 5). GSH, glutathione; NAA, N-acetyl aspartate.

**Figure 4 metabolites-14-00699-f004:**
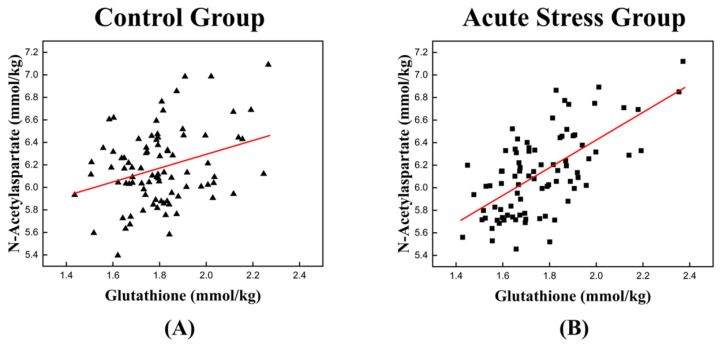
Scatter plot depicting the correlation between GSH and NAA levels in the left mouse hippocampus of (**A**) the control group and (**B**) the acute stress group. (**A**) A weak positive relationship is observed, suggesting that as the concentration of GSH increases, the levels of NAA also tend to increase. (**B**) A stronger positive correlation is observed in the acute stress group than in the control group. A higher correlation coefficient implies a stronger relationship between GSH and NAA under acute stress conditions. GSH, glutathione; NAA, N-acetyl aspartate. The red line represents the linear trend line, illustrating the relationship between GSH and NAA levels.

**Table 1 metabolites-14-00699-t001:** Mean concentrations (±standard error) of gamma-aminobutyric acid (GABA), glutamate (Glu), glutamine (Gln), glutathione (GSH), and N-acetyl aspartate (NAA) in the left hippocampus measured at different time points in the control and acute stress groups.

Mean Concentrations of Metabolites in the Left Hippocampus (Standard Error)									
Metabolites	Groups	Time Point 1	Time Point 2	Time Point 3	Time Point 4	Time Point 5	Time Point 6	Time Point 7	Time Point 8
GABA	Control	2.050 (0.065)	1.980 (0.060)	1.907 (0.053)	1.939 (0.097)	1.843 (0.100)	1.856 (0.073)	1.761 (0.086)	1.805 (0.121)
	Acute stress	1.986 (0.071)	1.941 (0.049)	1.817 (0.048)	1.819 (0.046)	1.916 (0.078)	1.785 (0.060)	1.902 (0.067)	1.793 (0.089)
Gln	Control	2.633 (0.215)	2.634 (0.193)	2.735 (0.215)	2.719 (0.202)	2.850 (0.221)	2.928 (0.074)	2.865 (0.180)	2.861 (0.221)
	Acute stress	2.332 (0.067)	2.303 (0.078)	2.390 (0.087)	2.409 (0.071)	2.577 (0.072)	2.587 (0.072)	2.653 (0.067)	2.699 (0.063)
Glu	Control	8.275 (0.139)	8.129 (0.162)	7.912 (0.110)	7.797 (0.127)	7.587 (0.169)	7.554 (0.110)	7.595 (0.149)	7.394 (0.167)
	Acute stress	8.582 (0.112)	8.439 (0.085)	8.200 (0.101)	8.041 (0.080)	8.145 (0.151)	7.809 (0.204)	7.768 (0.211)	7.668 (0.162)
GSH	Control	1.670 (0.032)	1.764 (0.056)	1.830 (0.037)	1.795 (0.044)	1.829 (0.051)	1.872 (0.060)	1.820 (0.060)	1.895 (0.056)
	Acute stress	1.661 (0.057)	1.677 (0.043)	1.704 (0.041)	1.695 (0.057)	1.790 (0.050)	1.866 (0.060)	1.850 (0.049)	1.828 (0.068)
NAA	Control	6.149 (0.105)	6.097 (0.098)	6.074 (0.079)	6.098 (0.092)	6.205 (0.128)	6.203 (0.110)	6.348 (0.137)	6.478 (0.166)
	Acute stress	6.015 (0.010)	5.964 (0.068)	5.972 (0.075)	6.042 (0.076)	6.260 (0.126)	6.252 (0.137)	6.266 (0.124)	6.248 (0.116)

**Table 2 metabolites-14-00699-t002:** Correlation coefficients between gamma-aminobutyric acid (GABA), glutamate (Glu), glutamine (Gln), glutathione (GSH), and N-acetyl aspartate (NAA) in the control and acute stress groups. * indicates statistical significance. ** indicates a statistically significant correlation coefficient of 0.5 or higher.

	Control Group			Acute Stress Group		
Pairs of Metabolites	Mean Correlation Coefficient	*p*-Value	Adjust *p*-Value	Mean Correlation Coefficient	*p*-Value	Adjust *p*-Value
GABA–Gln	0.190	1.000	1.000	−0.286	1.000	1.000
GABA–Glu	−0.429	0.012	0.060	0.429	0.227	0.454
GABA–GSH	0.024	1.000	1.000	−0.119	0.549	0.610
GABA–NAA	0.119	0.549	0.784	0.024	0.344	0.573
Gln–Glu	−0.143	0.227	0.378	−0.143	0.012	0.040 *
Gln–GSH	0.467	0.065	0.162	0.571	0.021	0.052
Gln–NAA	0.571	0.012	0.060	0.381	0.012	0.040 **
Glu–GSH	0.180	0.227	0.378	−0.024	0.549	0.610
Glu–NAA	−0.286	1.000	1.000	0.214	0.549	0.610
GSH–NAA	0.144	0.065	0.162	0.571	0.012	0.040 *

## Data Availability

The data are not publicly available due to ethical restrictions.

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
