# Peer review of "Effects of Acute Stress on Metabolic Interactions Related to the Tricarboxylic Acid (TCA) Cycle in the Left Hippocampus of Mice"

_metabolites, 2024, doi:10.3390/metabo14120699_

Round 1

Reviewer 1 Report

Comments and Suggestions for Authors

The paper " Impact of Acute Stress on Brain Metabolic Interactions in the Left Murine Hippocampus" presents new and interesting data using time-series proton magnetic resonance spectroscopy on changes in the tricarboxylic acid cycle in response to acute stress. Overall, the paper is of high interest because it allows for a new interpretation of the data related to positive correlations between tricarboxylic acids and acute stress phenomena.

The following comments arose during the review process:

 1. The Introduction contains objective information substantiating the problems of the article. It is recommended to provide more up-to-date references.

 2. In the Materials and Methods section, the authors should provide a more detailed protocol for calculating the reference signal. It is necessary to indicate the version of the FID-A software based on MATLAB.

3. The algorithm for calculating significant metabolic correlation coefficients in the MRS time series data is unclear. The authors are asked to provide more complete data for calculating the parameters.

 4. It is necessary to present the results of this work more fully and consistently. The logic of the authors and how the quantitative data were obtained is not always clear.

 5. The Discussion section is very interesting and informative. It is necessary to align the Results section and the Discussion section.

Comments on the Quality of English Language

English needs to be improved

Author Response

# Reviewer 1

The paper " Impact of Acute Stress on Brain Metabolic Interactions in the Left Murine Hippocampus" presents new and interesting data using time-series proton magnetic resonance spectroscopy on changes in the tricarboxylic acid cycle in response to acute stress. Overall, the paper is of high interest because it allows for a new interpretation of the data related to positive correlations between tricarboxylic acids and acute stress phenomena.

Dear reviewer,

Thank you for your comments and suggestions concerning our paper entitled “Impact of Acute Stress on Brain Metabolic Interactions in the Left Murine Hippocampus”. The revised part is marked in red in the manuscript and the response was explained point-by-point for each comment.

The following comments arose during the review process:

1. The Introduction contains objective information substantiating the problems of the article. It is recommended to provide more up-to-date references.

Response: We appreciate the valuable comment. In response, we revised the second paragraph of the Introduction (beginning on line 59), added references [14] and [15] to support the updated information, and rearranged the references appropriately throughout the sentences.

  1. Hertz, Leif. "Astrocytic energy metabolism and glutamate formation—relevance for 13C-NMR spectroscopy and importance of cytosolic/mitochondrial trafficking." Magnetic resonance imaging 29.10 (2011): 1319-1329.
  2. Clark, Joseph F., et al. "N-acetylaspartate as a reservoir for glutamate." Medical hypotheses 67.3 (2006): 506-512.

2. In the Materials and Methods section, the authors should provide a more detailed protocol for calculating the reference signal. It is necessary to indicate the version of the FID-A software based on MATLAB.

Response: We thank the reviewer for this suggestion. As recommended in the revised texts in lines 149-156.

Updated text:
The reference signal was selected using the mean squared error, which had the smallest difference between each 512 FID signal. FID-A software [Version 1.2], which is based on MATLAB [R2020a, Natick, MA, USA: MathWorks Inc.], was used to perform the phase and frequency shifts between the reference FID signal and other FID signals [28,29]. After performing phase and frequency correction, 512 FID data were divided into two 256 FID data points to improve the temporal resolution of brain metabolite fluctuations Figure 1(B). Finally, 8 time points MRS data were used per group (control and acute stress).

3. The algorithm for calculating significant metabolic correlation coefficients in the MRS time series data is unclear. The authors are asked to provide more complete data for calculating the parameters.

Response: We thank the reviewer for this suggestion. Figures 1 and 2 have been updated to visually represent the workflow of the correlation coefficient calculation and to clarify the roles of key parameters in the analysis. These visualizations now include the data preprocessing steps, parameter settings, and correlation calculation method, which we hope provides a clearer understanding of our approach.

Additionally, we revised the method section to clarify calculating the Spearman correlation coefficient using time-series MRS data, as well as the statistical analysis process, and have included additional references. Specifically, we detailed the preprocessing steps for time-series data, the algorithm used for correlation coefficient calculations, and the statistical approach for assessing significance.

Updated text:
2.4. Statistical Analysis

The Spearman correlation coefficients between brain metabolites were calculated using the time-series MRS data for each mouse. For each group, a non-parametric one-tailed sign-rank test was performed on the metabolic correlation coefficients to evaluate their significance. To enhance the reliability of the results, False Discovery Rate (FDR) correction was applied to the p values obtained from the sign-rank test for each group. To select pairs of metabolites with strong associations, we chose metabolite pairs with a correlation coefficient greater than 0.5. Metabolite pairs with correlation coefficients having adjusted p-values greater than 0.05 were considered statistically significant [25]. Figure 2 illustrates the method for calculating metabolic correlation matrices for each group and demonstrates how the significant Spearman correlation coefficients between brain metabolites are compared between control and acute stress group.

4. It is necessary to present the results of this work more fully and consistently. The logic of the authors and how the quantitative data were obtained is not always clear.

Response: We thank the reviewer for this suggestion. We calculated the levels of the analyzed brain metabolites and their mean and standard errors in the Results section and included a table to present these values.

Updated text: Table 1 represents the mean concentrations (± standard error) of the metabolites (GABA, Gln, Glu, GSH, and NAA) across eight time points for control and acute stress groups.

Table 1. Mean concentrations (± standard error) of gamma-aminobutyric acid (GABA), glutamate (Glu), glutamine (Gln), glutathione (GSH) and N-acetyl aspartate (NAA) in the left hippocampus across measured at different time points in the control and acute stress groups.

Average Concentrations of Metabolites in the Left Hippocampus (Standard Error)

Metabolites

Groups

Time point 1

Time point 2

Time point 3

Time point 4

Time point 5

Time point 6

Time point 7

Time point 8

GABA

Control

2.050(0.065)

1.980(0.060)

1.907(0.053)

1.939(0.097)

1.843(0.100)

1.856(0.073)

1.761(0.086)

1.805(0.121)

Acute stress

1.986(0.071)

1.941(0.049)

1.817(0.048)

1.819(0.046)

1.916(0.078)

1.785(0.060)

1.902(0.067)

1.793(0.089)

Gln

Control

2.633(0.215)

2.634(0.193)

2.735(0.215)

2.719(0.202)

2.850(0.221)

2.928(0.074)

2.865(0.180)

2.861(0.221)

Acute stress

2.332(0.067)

2.303(0.078)

2.390(0.087)

2.409(0.071)

2.577(0.072)

2.587(0.072)

2.653(0.067)

2.699(0.063)

Glu

Control

8.275(0.139)

8.129(0.162)

7.912(0.110)

7.797(0.127)

7.587(0.169)

7.554(0.110)

7.595(0.149)

7.394(0.167)

Acute stress

8.582(0.112)

8.439(0.085)

8.200(0.101)

8.041(0.080)

8.145(0.151)

7.809(0.204)

7.768(0.211)

7.668(0.162)

GSH

Control

1.670(0.032)

1.764(0.056)

1.830(0.037)

1.795(0.044)

1.829(0.051)

1.872(0.060)

1.820(0.060)

1.895(0.056)

Acute stress

1.661(0.057)

1.677(0.043)

1.704(0.041)

1.695(0.057)

1.790(0.050)

1.866(0.060)

1.850(0.049)

1.828(0.068)

NAA

Control

6.149(0.105)

6.097(0.098)

6.074(0.079)

6.098(0.092)

6.205(0.128)

6.203(0.110)

6.348(0.137)

6.478(0.166)

Acute stress

6.015(0.010)

5.964(0.068)

5.972(0.075)

6.042(0.076)

6.260(0.126)

6.252(0.137)

6.266(0.124)

6.248(0.116)

Control group

Acute stress group

Pairs of metabolites

Mean correlation

coefficient

p - value

Adjust

p - value

Mean correlation

coefficient

p - value

Adjust

p - value

GABA - Gln

0.190

1.000

1.000

-0.286

1.000

1.000

GABA - Glu

-0.429

0.012

0.060

0.429

0.227

0.454

GABA - GSH

0.024

1.000

1.000

-0.119

0.549

0.610

GABA -NAA

0.119

0.549

0.784

0.024

0.344

0.573

Gln - Glu

-0.143

0.227

0.378

-0.143

0.012

0.040*

Gln - GSH

0.467

0.065

0.162

0.571

0.021

0.052

Gln - NAA

0.571

0.012

0.060

0.381

0.012

 0.040**

Glu - GSH

0.180

0.227

0.378

-0.024

0.549

0.610

Glu - NAA

-0.286

1.000

1.000

0.214

0.549

0.610

GSH - NAA

0.144

0.065

0.162

0.571

0.012

0.040*

Table 2 presents the correlation coefficients and the adjusted p-values (at a significance level of 0.05) between metabolites related to the TCA cycle in both experimental groups. No significant correlations were observed in the control group. However, in the acute stress group, significant metabolic correlations were found for Gln-Glu, Gln-NAA, and GSH-NAA, with only the GSH-NAA pair exhibiting a high positive correlation coefficient greater than 0.5.

Table 2. Correlation coefficients between gamma-aminobutyric acid (GABA), glutamate (Glu), glutamine (Gln), glutathione (GSH) and N-acetyl aspartate (NAA) in the control and acute stress groups. * Indicates statistical significance. The high correlation between GSH and NAA in the acute stress group is illustrated in Figure 3, which depicts the temporal changes in the mean concentrations of GSH and NAA for both experimental groups. For GSH, the control group exhibited an approximately monotonic increase, ranging from 1.670 to 1.895 (Figure 3 (A)). In contrast, the acute stress group displayed a slight increase from 1.661 to 1.695 up to time point 4, followed by a significant rise to 1.828 after time point 4 (Figure 3 (C)). For NAA, the control group exhibited a slight decrease in mean concentration from 6.149 to 6.074 up to time point 3, followed by a monotonic increase to 6.478 after time point 3 (Figure 3 (B)). In the acute stress group, the mean concentration of NAA showed a pattern similar to that of the control group up to time point 4, with slightly lower values than the control group, although the difference was not statistically significant. From time point 5 onward, the mean remained nearly constant (Figure 4 (D)). Compared to the control group, the mean concentrations of GSH and NAA in the acute stress group exhibited similar temporal patterns (Figure (C) and Figure (D)). Until time point 4, there were little changes in the mean concentrations, and the values were lower than those of the control group. From time point 5 onward, the concentrations increased and became comparable to those of the control group, with little or no changes over time. Figure 4 shows the correlation between GSH and NAA in the mice hippocampus between the control and acute stress groups. The acute stress group showed a strong positive correlation between GSH and NAA, indicating that changes in GSH levels had a more pronounced impact on NAA levels compared to the control group.

5. The Discussion section is very interesting and informative. It is necessary to align the Results section and the Discussion section.

Response: We thank the reviewer for this suggestion. We updated the Discussion section to align with the Results by incorporating the supplementary Table and Figure. Additionally, we revised the Discussion to provide a more detailed interpretation of the key findings presented in the Results section, ensuring a logical connection between the data and their implications.

Specifically, we updated how the quantitative data from the Results section supports the interpretations and biological mechanisms discussed. We also ensured that each major point in the Results section is reflected and elaborated upon in the Discussion, with references to relevant data points, figures, and tables to improve the flow and consistency.

Reviewer 2 Report

Comments and Suggestions for Authors

Please, in line 48 and 50, correct the TCY to TCA.

Please, provide some more information (1-2 sentences) regarding the ''restrained stress''.

Please, make it clear in the text that you excecuted the experiment in hippocampus. It is mentioned more the ''brain'' samples than ''hippocampus '', that can lead to confusion.

Please, in Figure 3, include titles in the top part of the images, that will directly link the A and C with the Control group, and the B and D with the stressed group.

Please, provide an explanation of why you did not measure the N-acetylCysteine (NAC) in the different groups, as it is directly related to glutathione metabolism and stress response. It would be good to be included in the discussion (propably in the limitations).

Finally, please, explain how you could evaluate the stress between the different groups. It will be helpful, if you could measure the total ROS or measure any metabolite (other than the targeted GSH-related) that is direct indicator of stress (even ROS inducer).

Author Response

Reviewer 2

Dear reviewer,

Thank you for your comments and suggestions concerning our paper entitled “Impact of Acute Stress on Brain Metabolic Interactions in the Left Murine Hippocampus”. The revised part is marked in red in the manuscript and the response was explained point-by-point for each comment.

1. In line 48 and 50, correct the TCY to TCA.

Response: We apologize for this typo. As you suggested, we have corrected 'TCY' to 'TCA' and reviewed the entire manuscript for any other typos.

2. Provide some more information (1-2 sentences) regarding ‘restrained stress’.

Response: We thank the reviewer for this suggestion. As your comment we provided more information and details in line 108-119.

Updated text:

Mice in the acute stress group were exposed to restraint stress for 2 h in a 50 mL plastic conical tube (3 cm in diameter and 10 cm in length), following the stress protocol de-scribed in previous studies [23-27]. The plastic tube had holes near the body (3 mm in diameter) and the mouth (5 mm in diameter) of the mice to promote ventilation and airflow during the procedure. To ensure the mice were immobilized completely, the residual space in the tube was blocked and filled with material to pressurize and secure the body and head of the mouse. The restraint stress procedure was conducted in an air-conditioned room under background noise to maintain consistency in the experimental environment. Immediately after the 2-hour restraint stress protocol was completed, mice were used for time-series MRS data acquisition. Mice in the control group were not exposed to restraint stress and remained undisturbed in their cages until the beginning of data acquisition.

3. Make it clear in the text that you executed the experiment in hippocampus. It is mentioned more the ‘brain’ samples than ''hippocampus '', that can lead to confusion.

Response: We thank the reviewer for this suggestion. As your comment we have revised the text to specify that the experiment was conducted in the hippocampus. We have replaced references to 'brain' with 'hippocampus' where appropriate to avoid any confusion and ensure clarity regarding the sample location.

4. Figure 3, include titles in the top part of the images, that will directly link the A and C with the Control group, and the B and D with the stressed group.

Response: Thank you for this suggestion. We have updated Figure 3 by adding titles at the top of the images to clearly link panels.

(Figure in attached file)

Figure 3. Dynamic fluctuations of GSH and NAA in the control and acute stress groups over time. In the control group (A), GSH levels are initially higher compared to those in the acute stress group (C). However, GSH levels in the acute stress group begin to increase time point 5, eventually reaching levels similar to those of the control group. In the control group (B), NAA levels gradually increased over time, whereas in the acute stress group (D), NAA levels rapidly increased after time point 5. GSH, glutathione; NAA, N-acetyl aspartate.

5. Provide an explanation of why you did not measure the N-acetylCysteine (NAC) in the different groups, as it is directly related to glutathione metabolism and stress response. It would be good to be included in the discussion (propably in the limitations).

Response: Thank you for this important suggestion. We recognize that N-acetylcysteine (NAC) plays a critical role in glutathione metabolism and the stress response, acting as a precursor for glutathione synthesis and contributing to redox homeostasis. However, measuring NAC levels using 1H-MRS is challenging due to its low concentration and overlapping spectral peaks with other metabolites. For this reason, NAC was excluded from our analysis. We acknowledge this limitation and have added it to both the Discussion section and the limitations subsection, highlighting its potential impact on understanding glutathione metabolism and the stress response.

Updated text: Third, we did not measure N-acetylcysteine (NAC), a precursor of GSH, that is directly involved in stress response and glutathione metabolism because measurement of NAC is challenging due to low concentration and spectral overlap with other N-acetyl compounds, such as NAA [46]. NAC measurement could provide a more comprehensive understanding of the metabolic response to acute stress.

6. Explain how you could evaluate the stress between the different groups. It will be helpful, if you could measure the total ROS or measure any metabolite (other than the targeted GSH-related) that is direct indicator of stress (even ROS inducer).

Response: Thank you for this valuable question. In this study, we did not assess other stress-related markers, such as total ROS or oxidative stress indicators, which could serve as direct indicators of physiological stress and provide additional insights into the stress response between groups.

These markers would help clarify the extent of oxidative stress and its relationship to metabolic changes. However, our previous study demonstrated that the stress group showed a significant increase in glutamate levels, which we attributed to the effects of acute stress. Glutamate, an excitatory neurotransmitter, plays an important role in the acute stress response and is linked to glutathione metabolism [23,24]. While this study focused on GSH-related metabolites, the observed changes in glutamate further support the acute stress response in the stress group.

Updated text:

Fourth, we did not assess other stress-related markers, such as total ROS or oxidative stress indicators (e.g., malondialdehyde or superoxide dismutase), which provide the precursor of the stress response.

Reviewer 3 Report

Comments and Suggestions for Authors

Utilizing time series 1H-MRS, this study aims to investigate the effects of the acute stress response on brain metabolites and to elucidate the interactions among them. The authors analyzed five brain metabolites—glutamine (Gln), glutamate (Glu), glutathione (GSH), gamma-aminobutyric acid (GABA), and N-acetylaspartate (NAA)—which are involved in the glutamatergic and aspartate systems, closely related to the tricarboxylic acid (TCA) cycle. The results indicate that these metabolites may interact with one another following acute stress responses. However, there are several issues that the authors should address:1. The title of the manuscript is overly broad and does not accurately reflect the study and its findings.2. Regarding the interactions among metabolites, the results primarily focus on the correlation between GSH and NAA through statistical analysis, which falls short of adequately revealing the nature of their interactions.

Author Response

Reviewer 3

Dear reviewer,

Thank you for your comments and suggestions concerning our paper entitled “Impact of Acute Stress on Brain Metabolic Interactions in the Left Murine Hippocampus”. The revised part is marked in red in the manuscript and the response was explained point-by-point for each comment.

1) The title of the manuscript is overly broad and does not accurately reflect the study and its findings

Response: Thank you for this suggestion. We changed the article title to “Effects of Acute Stress on Metabolic Interactions Related to the Tricarboxylic Acid (TCA) Cycle in the Left Hippocampus of Mice” to better reflect the scope and findings of our study.

2) Regarding the interactions among metabolites, the results primarily focus on the correlation between GSH and NAA through statistical analysis, which falls short of adequately revealing the nature of their interactions.

Response: We appreciate the insightful comment regarding the interactions between GSH and NAA.

To address this, we significantly expanded the discussion in the revised manuscript and add more references.

Updated text: NAA is primarily synthesized in neurons which aspartate and acetyl-CoA combine in mitochondria to form NAA [36]. It plays an important role in neuronal energy metabolism, protein synthesis, and provides acetate for myelin synthesis. Following acute stress, NAA levels initially decrease, indicating a transient disruption in neuronal metabolism, and later increase, reflecting enhanced metabolic activity and neuroprotection to restore homeostasis [37,38]. The interaction between NAA and GSH is vital for maintaining mitochondrial function. While NAA supports mitochondrial energy production, GSH neutralizes ROS, preventing oxidative damage caused by stress [33-35]. This interplay ensures mitochondrial integrity and supports neuronal energy metabolism. Additionally, the TCA cycle within mitochondria oxidizes acetyl-CoA derived from carbohydrates, fats, and proteins to produce ATP, which is essential for neurotransmitter synthesis and cellular recovery. NAA synthesis depends on ATP, as aspartate and acetate are utilized to form NAA [39-43]. As shown in Figure 3, NAA levels reflected an adaptive neuroprotective response. Initially, NAA levels decreased, indicating a transient disruption in neuronal metabolism following acute stress exposure. Subsequently, NAA levels increased, suggesting enhanced metabolic activity and neuroprotection aimed to restore neuronal homeostasis [19,39-41]. This metabolic interaction depicts the critical roles of GSH, NAA, and the TCA cycle in supporting neuronal resilience. GSH mitigates oxidative stress by neutralizing ROS, while NAA ensures mitochondrial energy production and supplies acetate for essential processes such as myelin synthesis and neuronal repair. The TCA cycle sustains ATP production, which is necessary for neurotransmitter synthesis and cellular function. Together, these mechanisms underline the adaptive neuroprotective response of NAA in maintaining neuronal health under acute stress [43-49].

Round 2

Reviewer 1 Report

Comments and Suggestions for Authors

In the revised version of the article, the authors took into account all the comments made by the reviewer and improved the content of the manuscript many times. In the revised version, the manuscript can be recommended for publication in the journal Metabolitesю

Reviewer 3 Report

Comments and Suggestions for Authors

The authors has addressed my concerns.